# Visualizing and Understanding the Semantics of Embedding Spaces via Algebraic Formulae

## Abstract

Embeddings are a fundamental component of many modern machine learning and natural language processing models. Understanding them and visualizing them is essential for gathering insights about the information they capture and the behavior of the models. State of the art in analyzing embeddings consists in projecting them in two-dimensional planes without any interpretable semantics associated to the axes of the projection, which makes detailed analyses and comparison among multiple sets of embeddings challenging. In this work, we propose to use explicit axes defined as algebraic formulae over embeddings to project them into a lower dimensional, but semantically meaningful subspace, as a simple yet effective analysis and visualization methodology. This methodology assigns an interpretable semantics to the measures of variability and the axes of visualizations, allowing for both comparisons among different sets of embeddings and fine-grained inspection of the embedding spaces. We demonstrate the power of the proposed methodology through a series of case studies that make use of visualizations constructed around the underlying methodology and through a user study. The results show how the methodology is effective at providing more profound insights than classical projection methods and how it is widely applicable to many other use cases.

## 1 Introduction

Learning representations is an important part of modern machine learning and natural language processing research. Those representations are often real-valued vectors also called embeddings and are obtained both as byproducts of supervised learning or as the direct goal of unsupervised methods. Independently of how the embeddings are learned, there is much value in understanding what information they capture, how they relate to each other and how the data they are learned from influences them. A better understanding of the embedded space may lead to a better understanding of the data, of the problem and the behavior of the model, and may lead to critical insights in improving such models. Because of their high-dimensional nature, they are hard to visualize effectively, and the most adopted approach is to project them in a bi-dimensional space. Projections have a few shortcomings: 1) they may not preserve distance in the original space, 2) they are not comparable across models and 3) do not provide interpretable dimensions of variability to project to, preventing for more detailed analysis and understanding. For these reasons, there is value in mapping embeddings into a more specific, controllable and interpretable semantic space.

Principal Component Analysis (PCA) (Pearson, 1901) and t-Distributed Stochastic Neighbor Embedding (t-SNE) (van der Maaten & Hinton, 2008) are two projection techniques often used for visualizing embeddings in two dimensions, although other techniques can be used. PCA projects embeddings on a lower dimensional space that has the directions of the highest variance in the dataset as axes. Those dimensions do not carry any interpretable meaning, making interpretation difficult. By visualizing the first two dimensions of a PCA projection, the only insight obtainable is semantic relatedness (Budanitsky & Hirst, 2006) between points by observing their relative closeness and therefore topical clusters can be identified. The downside is that embeddings that end up being close in the projected space may not be close in the original embedding space and vice versa. Moreover, as the directions of highest variance are different from embedding space to embedding space, the projections are incompatible among different embeddings spaces, and this makes them not comparable, a common issue among dimensionality reduction techniques.

t-SNE, differently from PCA, optimizes a loss that encourages embeddings that are close in the original high-dimensional space to be close in the lower dimensional projection space. This helps in visualizing clusters better than with PCA, as t-SNE puts each point in the projected space so that distance in the original space with respect to its nearest neighbors is preserved as much as possible. Visualizations obtained in this way reflect more the original embedding space and topical clusters are more clearly distinguishable, but doesn't solve the issue of comparability of two different sets of embeddings, nor it solves the lack of interpretability of the axes and still doesn't allow for fine-grained inspection. Moreover, t-SNE is pretty sensible to hyperparameters, making it unclear how much the projection reflects the data.

In this paper, a new and simple method to inspect, explore and debug embedding spaces at a fine-grained level is proposed. It consists in defining explicitly the axes of projection through formulae in vector algebra over the embeddings themselves. Explicit axis definition gives an interpretable and fine-grained semantics to the axes of projection. Defining axes explicitly makes it possible to analyze in a detailed way how embeddings relate to each other with respect to interpretable dimensions of variability, as carefully crafted formulas can map (to a certain extent) to semantically meaningful portions of the learned spaces. The explicit axes definition also allows for comparing of embeddings obtained from different datasets, as long as they have common labels.

We demonstrate three visualizations for analyzing subspaces of interest of embedding spaces and a set of example case studies including bias detection, polysemy analysis and fine-grained embedding analysis. Additional tasks that may be performed using the proposed methodology and visualization are diachronic analysis and analysis of representations learned from graphs and knowledge bases. The proposed visualizations can moreover be used for debugging purposes and in general for obtaining a better understanding of the embedding spaces learned by different models and representation learning approaches. We are releasing an open-source [1] interactive tool that implements the proposed visualizations, in order to enable researchers in the fields of machine learning, computational linguistics, natural language processing, social sciences and digital humanities to perform exploratory analysis and better understand the semantics of their embeddings.

The main contribution of this work lies in the use of explicit user-defined algebraic formulae as axes for projecting embedding spaces into semantically-meaningful subspaces that when visualized provide interpretable axes. We show how this methodology can be widely used through a series of case studies on well known models and data and we furthermore validate the how the visualizations are more interpretable through a user study.

## 2 RELATED WORK

### 2.1 EMBEDDING METHODS AND APPLICATIONS

Several methods for learning embeddings from symbolic data have been recently proposed (Pennington et al., 2014; Mikolov et al., 2013; Mnih & Kavukcuoglu, 2013; Lebret & Collobert, 2014; Ji et al., 2016; Rudolph et al., 2016; Nickel et al., 2016). The learned representations have been used for a variety of tasks like recommendation (Barkan & Koenigstein, 2016), link prediction on graphs (Grover & Leskovec, 2016), discovery of drug-drug interaction (Abdelaziz et al., 2017) and many more. In particular, positive results in learning embeddings for words using a surrogate prediction task (Mikolov et al., 2013) started the resurgence of interest in those methods, while a substantial body of research from the distributional semantics community using count and matrix factorization based methods (Deerwester et al., 1990; Baroni & Lenci, 2010; Kanerva et al., 2000; Levy & Goldberg, 2014; Biemann & Riedl, 2013; Gabrilovich & Markovitch, 2007) was previously developed. Refer to Lenci (2018) for a comprehensive overview.

### 2.2 EMBEDDING VISUALIZATION

In their recent paper, Heimerl & Gleicher (2018) extracted a list of routinely conducted tasks where embeddings are employed in visual analytics for NLP, such as *compare concepts*, *finding analogies*, and *predict contexts*. iVisClustering (Lee et al., 2012) represents topic clusters as their most

---

[1]The tool will be made available after the review period to preserve double-blindness

representative keywords and displays them as a 2D scatter plot and a set of linked visualization components supporting interactively constructing topic hierarchies. ConceptVector (Park et al., 2018) makes use of multiple keyword sets to encode the relevance scores of documents and topics: positive words, negative words, and irrelevant words. It allows users to select and build a concept iteratively. Liu et al. (2018) display pairs of analogous words obtained through analogy by projecting them on a 2D plane obtained through a PCA and an SVM to find the plane that separates words on the two sides of the analogy. Besides word embedding, textual visualization has been used to understand topic modeling (Chuang et al., 2012) and how topic models evolve over time (Havre et al., 2002). Compared to literature, our work allows more fine-grained control over the conceptual axes and the filtering logic, e.g., allowing users to define concept based on explicit algebraic formulae beyond single keywords (Section 3), metadata based filtering, as well as multidimensional and multi-data source comparison beyond the common 2D scatter plot view. (Sec 4)

## 3 METHODOLOGY

The basic insight of this work is that goal-oriented inspection of embedding spaces can be defined in terms of items and dimensions of variability. For instance, if the goal is to discover if a dataset (and by consequence an embedding model trained on it) includes gender bias, a user may define professions as specific items of interest and analyze how they are distributed among the concept of "male" and "female", the two dimensions of variability. We use this as a running example in this section, while in the next section we present how to turn goal definitions into visualizations.

The dimensions of variability are defined as algebraic formulae that use embedding labels as atoms. Algebraic formulae are a composition vector math operators (e.g., add, sub, mul) to be applied on vectors (referenced by their label in the data, i.e. the vector of "apple" is obtained by using using the keyword "apple" in the formula). They are used as the axes of a subspace of the entire embedding space and can be interpreted as concepts. In our example we can define two axes $a_{male} = man$ and $a_{female} = woman$. These are the most simple formulae as they are made of only one literal, but any formula using algebraic operation can be used instead. For instance $a_{male} = man + him$ and $a_{female} = woman + her$ could be used instead. Defining axes explicitly as algebraic formulae gives an interpretable semantics to the dimensions of variability and by consequence to the axes of the visualization. To project on the axes, different distance and similarity measures can be used (euclidean distance, correlation, dot product), in particular we will use cosine similarity in the remaining of the paper, defined as $cossim(\mathbf{a}, \mathbf{b}) = \frac{\mathbf{a} \cdot \mathbf{b}}{\|\mathbf{a}\|\|\mathbf{b}\|}$

The items of the goal are a set defined by extention ($items = \{item_1, \ldots, item_n\}$) or by intention (with rules). The rules use items' embeddings or items' metadata, like word frequencies, parts of speech, sentiment or categories the label belongs to. Intentions identify a semantically coherent region of the embedding space through logical formulae.

Rules using item's embeddings are defined as $r_e = \langle d, e, c, v \rangle$ where $d$ is a distance or similarity function, $e$ is an algebraic formula that uses embeddings names as atoms and is resolved into a vector, $c \in \{<, \leq, =, \geq, >\}$, $v$ is a numeric value. They can be used, for instance, to select all the items that have a $d = cosinesimilarity$ with respect to $e = job + profession$ that is $c = \geq$ then $v = 0.5$.

Rules using item's metadata instead use typed metadata associated with each item. An item can have categorical fields (e.g., words can be stop-words or not), set fields (e.g., the parts of speech a word can belongs to) and numerical fields (e.g., unigram frequencies in a corpus) associated with it. Rules are be defined as inclusion in a set: $r_m = i_{cat} \cap set \neq \emptyset$ where $i_{cat}$ is the set of categories associated with an item, containing only one element in the case of categorical fields or multiple values in the case of set fields, while for numerical fields they are defined as ranges.

Following on with our example, we can select some professions like "doctor", "nurse", "teacher" and "astronaut" as our items, or we can define the items of interest as the set of all words in the embedding space that are close to the word "profession" (e.g., cosine similarity greater than 0.7), that are not too frequent (inside a range of frequency from 100 to 1000) and that are not stop-words.

## 4 Visualizations

Goals defined in terms of dimensions of variability and items identify a subspace of the entire embedding space to visualize and the best way to visualize it depends on some characteristics of the goal.

In the case of few dimensions of variability (one to three) and potentially many items of interest, like the ones obtained by an empty set of rules, a Cartesian view is ideal, where each axis is the vector obtained by evaluating the algebraic formula it is associated with and the coordinates displayed are similarities or distances of the items with respect to each axis. An example of a bi-dimensional Cartesian view is depicted in the left side of Figure 1.

In the case where the goal is defined in terms of many dimensions of variability, the Cartesian view can't be used, and a polar view is preferred. By visualizing each dimension of variability in circle, the polar view can visualize many more axes, but it is limited in the number of items it can display, as each item will be displayed as a polygon with each vertex lying on the axis defined for each dimension of variability and many overlapping polygons make the visualization cluttered. An example of a five-dimensional polar view is depicted in Figure 5.

The use of explicit axes allows for straightforward and interpretable comparison of different embedding spaces. For instance, embeddings trained on different corpora or on the same corpora but with different models. The only requirement for embedding spaces to be comparable is that they contain embeddings for all labels present in the formulae defining the axes. Moreover, embeddings in the two spaces do not need to be of the same dimension. Items will now have two sets of coordinates, one for each embedding space, thus they will be displayed as lines. Short lines are interpreted as items being embedded similarly in the subspaces defined by the axes in both original embedding spaces, while long lines can be interpreted as really different locations in the subspaces, and their direction gives insight on how items shift in the two subspaces. Those two embedding spaces could be, for instance, embeddings trained on a clean corpus like Wikipedia as opposed to a noisy corpus like tweets from Twitter, or the two corpora could be two different time slices of the same corpus, in order to compare how words changed over time. The right side of Figure 1 shows an example of how to use the Cartesian comparison view to compare two datasets.

## 5 Case Studies

The methodology and visualizations can be used fruitfully in many analysis tasks in linguistics, digital humanities, in social studies based on empirical methods, and can also be used by researchers in computational linguistics and machine learning to inspect, debug and ultimately better understand the representations learned by their models. Here few of those use cases are presented, but the methodology is flexible enough to allow many other unforeseen uses. For those tasks, we used 50-dimensional publicly available GloVe (Pennington et al., 2014) embeddings trained on a corpus obtained concatenating a 2014 dump of Wikipedia and Gigaword 5 containing 6 billion tokens (for short *Wikipedia*) and a set of 2 billion tweets containing 27 billion tokens (for short *Twitter*).

### 5.1 Bias detection

The task of bias detection is to identify, and in some cases correct for, bias in data that is reflected in the embeddings trained on such data. Studies have shown how embeddings incorporate gender and ethnic biases (Garg et al. (2018); Bolukbasi et al. (2016); Islam et al. (2017)), while other studies focused on warping spaces in order to de-bias the resulting embeddings (Bolukbasi et al. (2016); Zhao et al. (2017)). We show how our proposed methodology can help visualize biases.

To visualize gender bias with respect to professions, the goal is defined with the formulae $avg(he, him)$ and $avg(she, her)$ as two dimensions of variability, in a similar vein to Garg et al. (2018). A subset of the professions used by Bolukbasi et al. (2016) is selected as items and cosine similarity is adopted as the measure for the projection. The Cartesian view visualizing *Wikipedia* embeddings is shown in the left side of Figure 1. *Nurse*, *dancer*, and *maid* are the professions that end up closer to the "female" axis, while *boss*, *captain*, and *commander* end up closer to the "male" axis.

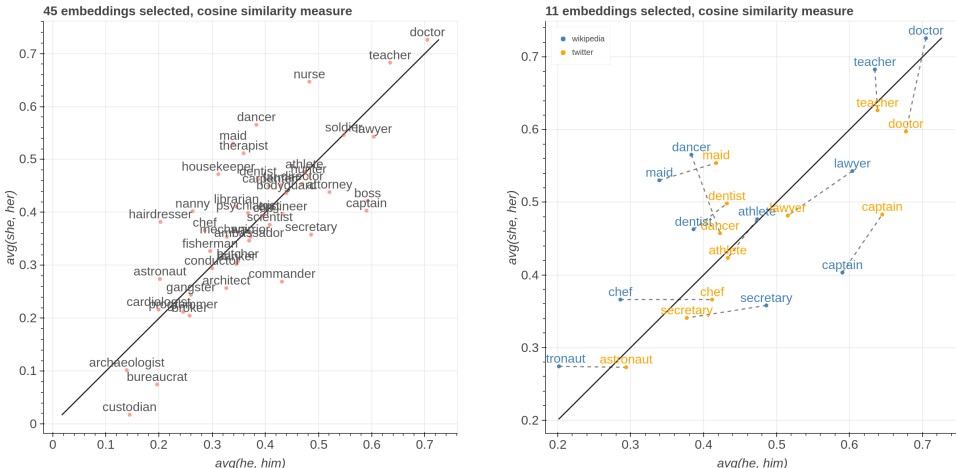

Figure 1: On the left we show professions plotted on "male" and "female" axes in $Wikipedia$ embeddings. On the right we show thier comparison in $Wikipedia$ and $Twitter$ datasets.

The Cartesian comparison view comparing the embeddings trained on *Wikipedia* and *Twitter* is shown in the right side of Figure 1. Only the embeddings with a line length above $0.05$ are displayed. The most interesting words in this visualization are the ones that shift the most in the direction of negative slope. In this case, are *chef* and *doctor* are closer to the "male" axis in *Twitter* than in *Wikipedia*, while *dancer* and *secretary* are closer to the bisector in *Twitter* than in *Wikipedia*.

Additional analysis of how words tend to shift in the two embedding spaces would be needed in order to derive provable conclusions about the significance of the shift, for instance through a permutation test with respect to all possible pairs, but the visualization can help inform the most promising words to perform the test on.

## 5.2 POLYSEMY ANALYSIS

Embedding methods conflate different meanings of a word into the same vector A few methods have been proposed to obtain more fine-grained representations by clustering contexts and representing words with multiple vectors (Huang et al., 2012; Neelakantan et al., 2014), but widely used pre-trained GloVe vectors still conflate different meanings in the same embedding.

Widdows (2003) showed how using a binary orthonormalization operator that has ties with the quantum logic *not* operator it is possible to remove from the embedding of a polysemous word part of the conflated meaning. The authors define the operator $nqnot(a,b) = a - \frac{a \cdot b}{|b|^2}b$ and we show with a comparison plot how it can help distinguish the different meanings of a word.

For illustrative purposes we choose the same polysemous word used by Widdows (2003), *suit*, and use the $nqnot$ operator to orthonormalize with respect to *lawsuit* and *dress*, the two main meanings used as dimensions of variability. The items in our goal are the 20000 most frequent words in the *Wikipedia* embedding space removing stop-words. In the top of Figure 2 we show the overall plot and we zoom on the items that are closer to each axis. Words closer to the axis negating *lawsuit* are all related to dresses and the act of wearing something, while words closer to the axis negating *dress* are related to law.

We chose another polysemous word, *apple*, and orthonornalized with respect to *fruit* and *computer*. In the bottom of Figure 2 words that have a higher similarity with respect to the first axis are all tech related, while the ones that have a higher similarity with respect to the second axis are mostly other fruits or food. Both examples confirm the ability of the $nqnot$ operator to disentangle multiple meanings from polysemous embeddings and show how the proposed visualizations are able to show it clearly.

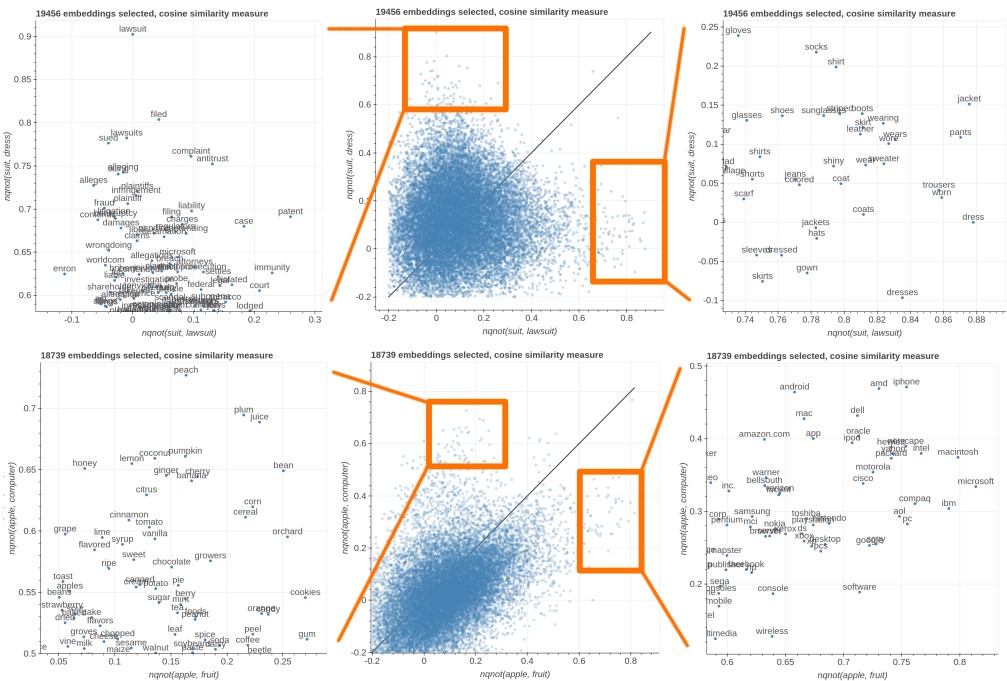

Figure 2: In the top a plot of embeddings in *Wikipedia* with *suit* negated with respect to *lawsuit* and *dress* respectively as axes. In the bottom a plot of *apple* negated with respect to *fruit* and *computer*.

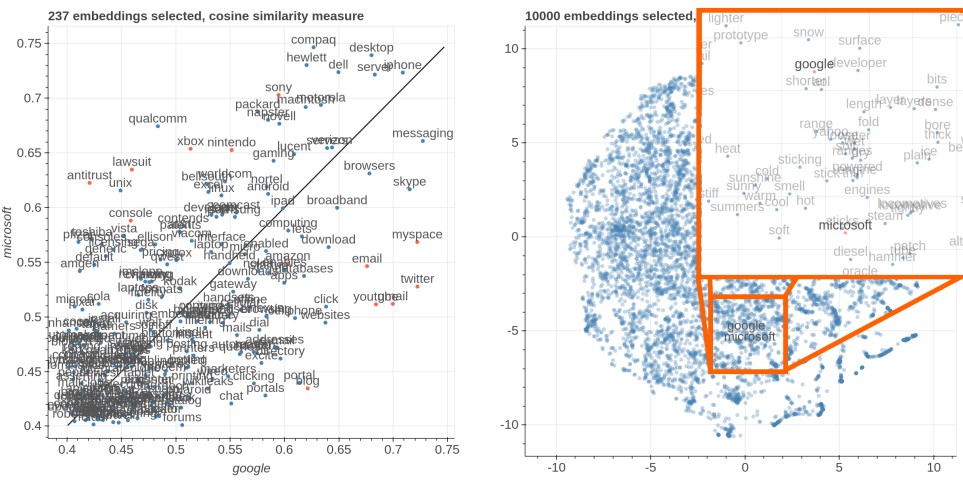

Figure 3: The left figure is a fine-grained comparison of the subspace on the axis *google* and *microsoft* in *Wikipedia*, the right one is the *t-SNE* conterpart.

## 5.3 FINE-GRAINED EMBEDDING ANALYSIS

We consider embeddings that are close in the embedding space to be semantically related, but even close embeddings may have nuances that distinguish them. When projecting in two dimensions through PCA or t-SNE we are conflating a multidimensional notion of similarity to a bi-dimensional one, loosing the fine grained distinctions among different embeddings. The Cartesian view allows for a more fine-grained visualization of similarities and differences among embeddings that emphasizes nuances that could go otherwise unnoticed.

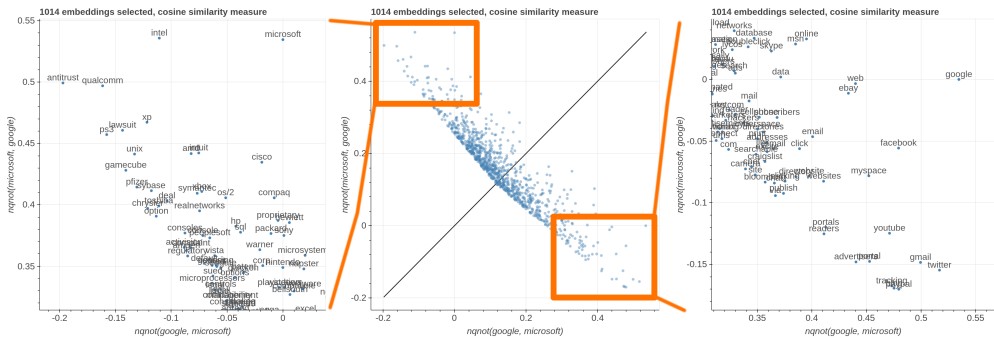

Figure 4: Fine-grained comparison of the subspace on the axis $nqnot(google, microsoft)$ and $nqnot(microsoft, google)$ in *Wikipedia*.

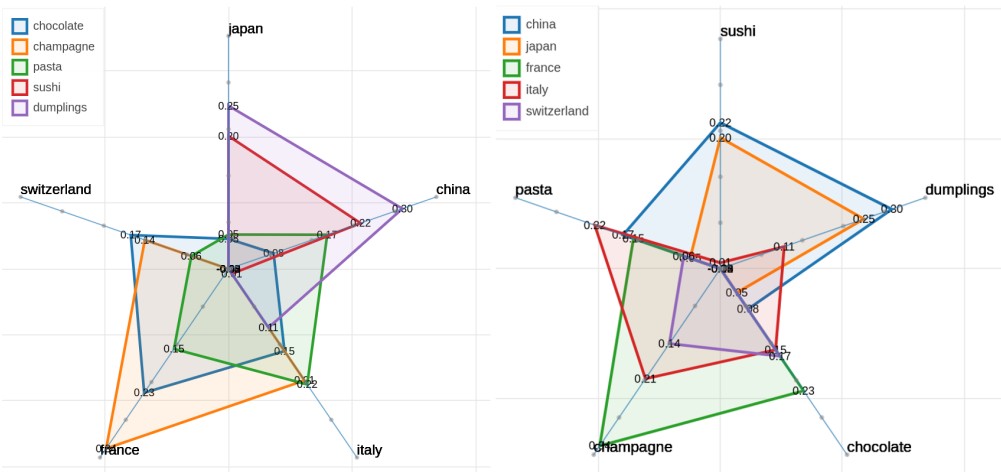

Figure 5: Two polar views of countries and foods.

To demonstrate this capability we select as dimensions of variability formulae made of just single words that are in close vicinity to each other in the *Wikipedia* embedding space: *google* and *microsoft*, as *google* is the closest word to *microsoft* and *microsoft* is the 3rd closest word to *google*. As items we pick the 30000 most frequent words removing stop-words and the 500 most frequent words (as they are too generic) and keeping only the words that have a cosine similarity of at least $0.4$ with both *google* and *microsoft* while having a cosine similarity below $0.75$ with respect to the formula $google + microsoft$, as we are interested in the most polarized words.

The left side of Figure 3 shows how even if those embeddings are close to each other it is easy to identify peculiar words (highlighted with red dots). The ones that relate to web companies and services (*twitter*, *youtube*, *myspace*) are much closer to the *google* axis. Words related to both legal issues (*lawsuit*, *antitrust*) and videogames (*ps3*, *nintendo*, *xbox*) and traditional IT companies are closer to the *microsoft* axis.

In Figure 4 we the same words using *google* and *microsoft* orthonormalized with respect to each other as axes. The top left and the bottom right corners are the most interesting ones, as they contain terms that are related to one word after having negated the other. The pattern that emerges is similar to the one highlighted in the left side of Figure 3, but now also operating systems terms (*unix*, *os/2*) appear in the *microsoft* corner, while *advertisement* and *tracking* appear in the *google* corner.

For contrast, the t-SNE projection is shown in the right side of Figure 3: it is hard to appreciate the similarities and differences among those embeddings other than seeing them being close in the projected space. This confirms on one hand that the notion of similarity between terms in an

| Measure | Projection × Task | | | Projection × Obfuscation | | |
|---------|-------|-----------|---------|-------|-----------|---------|
| | Factor | $F_{(1,91)}$ | p-value | Factor | $F_{(1,91)}$ | p-value |
| Accuracy | Projection | 46.11 | 0.000*** | Projection | 57.73 | 0.000*** |
| | Task | 1.709 | 0.194 | Obfuscation | 23.93 | 0.000*** |
| | Projection × Task | 3.452 | 0.066 | Projection × Obfuscation | 5.731 | 0.019* |
| Speed | Projection | 0.881 | 0.350 | Projection | 0.808 | 0.371 |
| | Task | 0.752 | 0.785 | Obfuscation | 5.901 | 0.017* |
| | Projection × Task | 2.899 | 0.092 | Projection × Obfuscation | 1.369 | 0.245 |

Table 1: Two-way ANOVA analyses of Task (Commonality vs. Polarization) and Obfuscation (Obfuscated vs. Non-obfuscated) over Projection (Explicit Formulae vs. t-SNE).

embedding space hides many nuances that are captured in those representations and on the other hand that the proposed methodology enables for a more detailed inspection of the embedded space.

Multi-dimensional similarity nuances can be visualized using the polar view. In Figure 5 we show an example of how to visualize a small number of items on more than two axes, specifically five food-related items compared over five countries axes. The most typical food from a specific country is the closest to the country axis, with *sushi* being predominantly close to *Japan* and *China*, *dumplings* being close to both Asian countries and *Italy*, *pasta* being predominantly closer to *Italy*'s axis, *chocolate* being close to European countries and *champagne* being closer to *France* and *Italy*. This same approach could be used also for bias detection where the axes are concepts capturing the notion of ethnicity and items could be adjectives, or the two could be swapped, depending.

## 6 USER STUDY

We conducted a series of user studies to quantify the effectiveness of the proposed method. The goal is to find out if and how visualizations using user-defined semantically meaningful algebraic formulae as their axes help users achieve their analysis goals. What we are not testing for is the quality of projection itself, as in PCA AND t-SNE the projection axes are obtained algorithmically, while in our case they are explicitly defined by the user. We formalized the research questions as: Q1) Does Explicit Formulae outperform t-SNE in goal-oriented tasks? Q2) Can Explicit Formulae reduce time to complete goal-oriented tasks wrt. t-SNE? Q3) Which visualization do users prefer?

To answer these questions we invited twelve subjects among data scientists and machine learning researchers, all acquainted with interpreting dimensionality reduction results. We defined two types of tasks, namely Commonality and Polarization, in which subjects were given a visualization together with a pair of words (used as axes in Explicit Formulae or highlighted with a big font and red dot in case of t-SNE). We asked the subjects to identify either common or polarized words w.r.t. the two provided ones. The provided pairs were: banana & strawberry, google & microsoft, nerd & geek, book & magazine. The test subjects were given a list of eight questions, four per task type, and their proposed lists of five words are compared with a gold standard provided by a committee of two computational linguistics experts. The tasks are fully randomized within the subject to prevent from learning effects. In addition, we obfuscated half of our questions by replacing the words with a random numeric ID to prevent prior knowledge from affecting the judgment. We employed three measures: *accuracy* in which we calculate the number of words provided by the subjects that are present in the gold standard set, *speed* recording the amount of time users spend to answer the questions normalized by the number of words (commonality: 1, polarization: 2), and we also collected an overall *preference* for either visualizations.

As reported in Table 1, two-way ANOVA tests revealed significant differences in Accuracy for the factor of Projection (Explicit Formulae ($\mu = 2.02$, $\sigma = 1.40$) and t-SNE ($\mu = 0.50$, $\sigma = 0.71$)) against both Task ($F_{1,91} = 46.11$, $p = 1.078 \times 10^{-9}$) and Obfuscation ($F_{1,91} = 57.73$, $p = 2.446 \times 10^{-11}$), which is a strong indicator that the proposed Explicit Formulae method outperforms t-SNE in terms of accuracy in both Commonality and Polarization tasks. We also observed significant differences ($F_{1,91} = 23.93$, $p = 4.228 \times 10^{-6}$) in Obfuscation: subjects tend to have better accuracy when the words are not obfuscated ($\mu = 1.75$, $\sigma = 1.55$ vs. $\mu = 0.77$, $\sigma = 0.88$

when obfuscated), but are significantly slower ($F_{1,91} = 5.901$, $p = 0.017$). We run post-hoc t-tests that confirmed how accuracy of Explicit Formulae on Non-obfuscated is better than Obfuscated ($t = 4.172$, $p < 0.0001$), which in turn is better that t-SNE Non-obfuscated ($t = 2.137$, $p = 0.0190$), which is better than t-SNE Obfuscated ($t = 2.563$, $p = 0.007$). One explanation is that the subjects relied on both the visualization and linguistic knowledge to perform the task, but the fact that Explicit Formulae Obfuscated is still better than t-SNE Non-obfuscated suggests that Explicit Formulae, even with obfuscated labels, is consistently more reliable than t-SNE. Concerning Speed, we did not observe signs that the subjects performed faster with Explicit Formulae comparing to t-SNE. Concerning Preference, nine out of all twelve (75%) subjects chose Explicit Formulae over t-SNE, while the rest three prefers t-SNE because of familiarity, indicating there is still non-negligible learning curve of our proposed methods.

In conclusion, our answers to the research questions are that (Q1) Explicit Formulae leads to better ac curacy in goal-oriented tasks, (Q2) there is no significant difference between the two techniques in terms of speed and (Q3) users prefer Explicit Formulae over t-SNE.

## 7    CONCLUSIONS

We presented a simple methodology for projecting embeddings into lower-dimensional semantically-meaningful subspaces through explicit vector algebra formulae operating on the embedding themselves. Classical projection methods are useful to gather on overall coarse-grained view of the embedding space and how embeddings cluster, but we showed how our approach allows goal-oriented analyses with more fine-grained comparison and enables cross-dataset comparison through a series of case studies and a user study. This is possible thanks to the ability of the proposed methodology to assign an explicit semantics to the measures of variability used as axes of the visualization that in turns makes them interpretable and widely applicable to many use cases in computational linguistics, natural language processing, machine learning, social sciences and digital humanities.

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

# A    APPENDIX

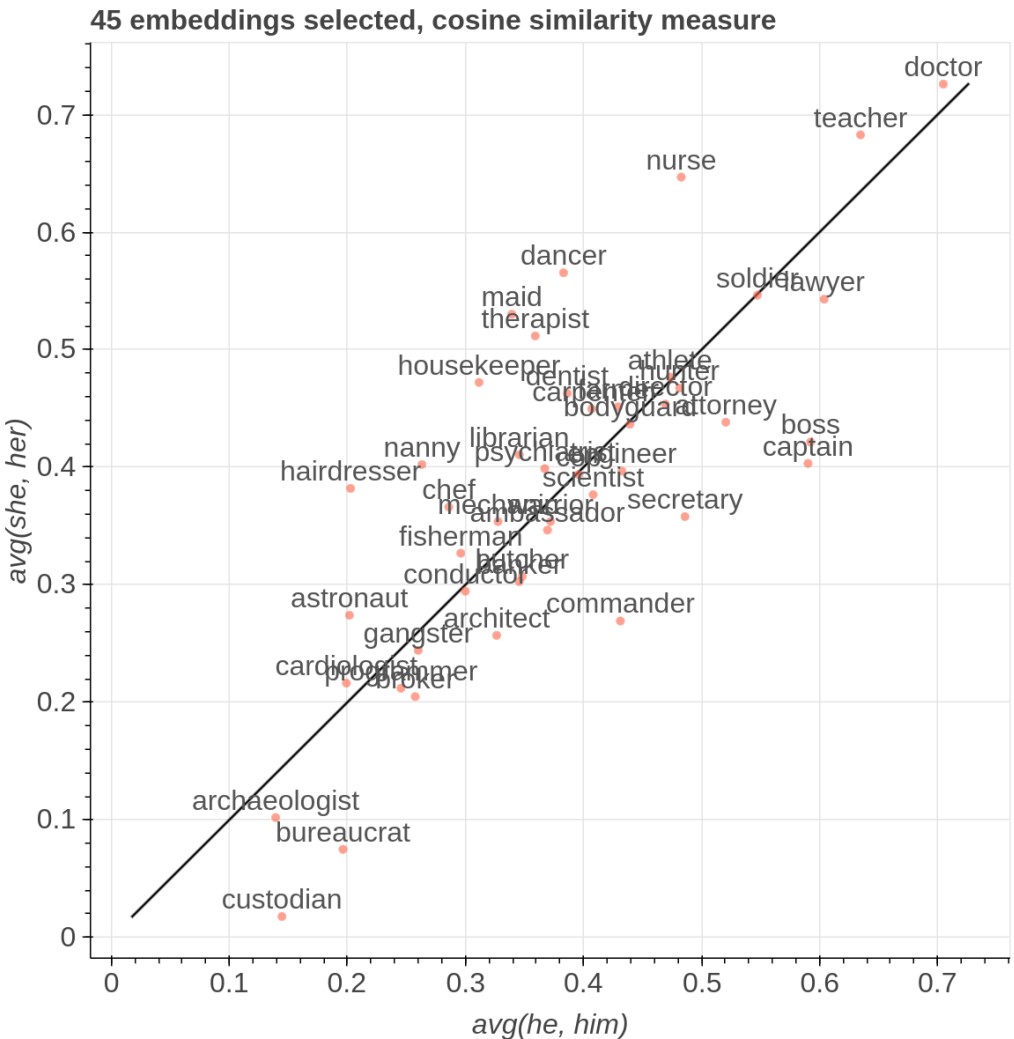

Figure 6: Professions plotted on "male" and "female" axes in $Wikipedia$ embeddings.

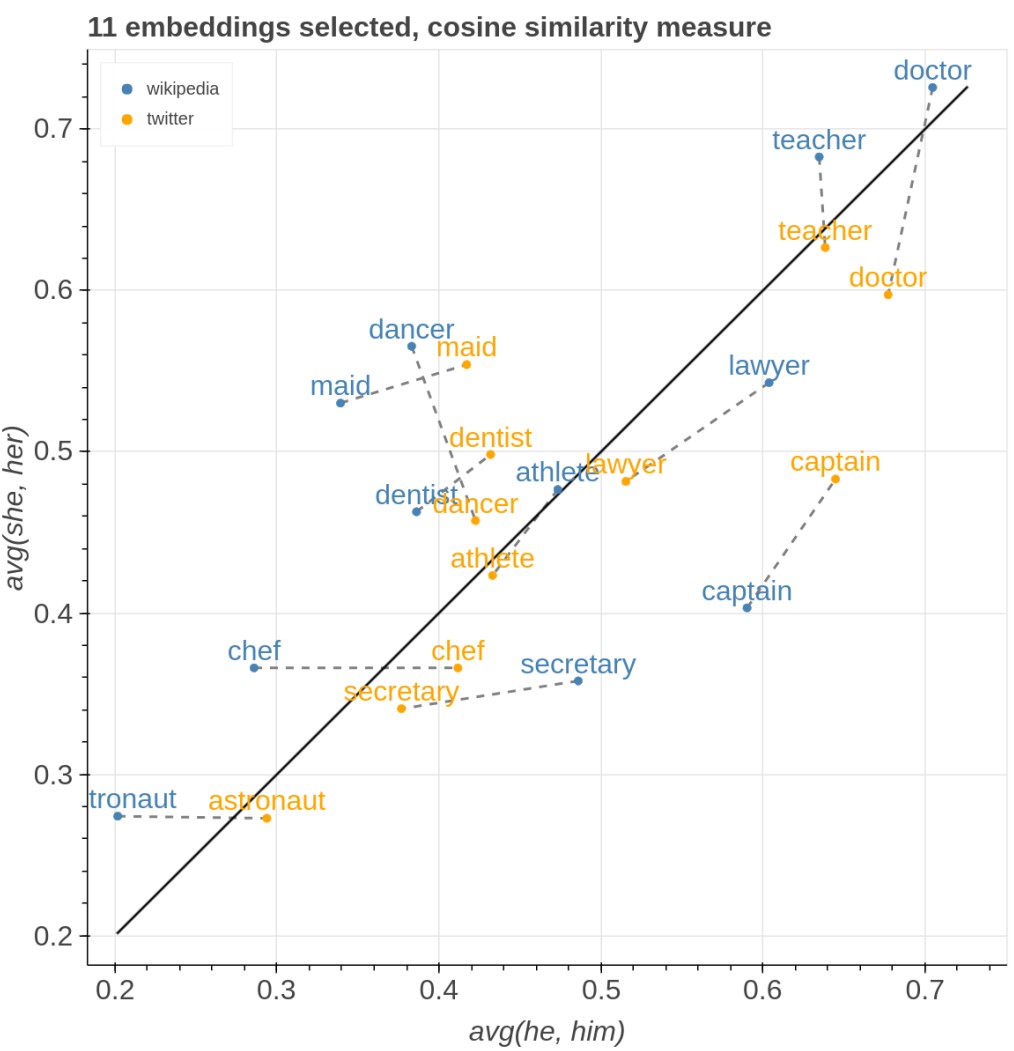

Figure 7: Professions plotted on "male" and "female" axes in $Wikipedia$ and $Twitter$ embeddings.

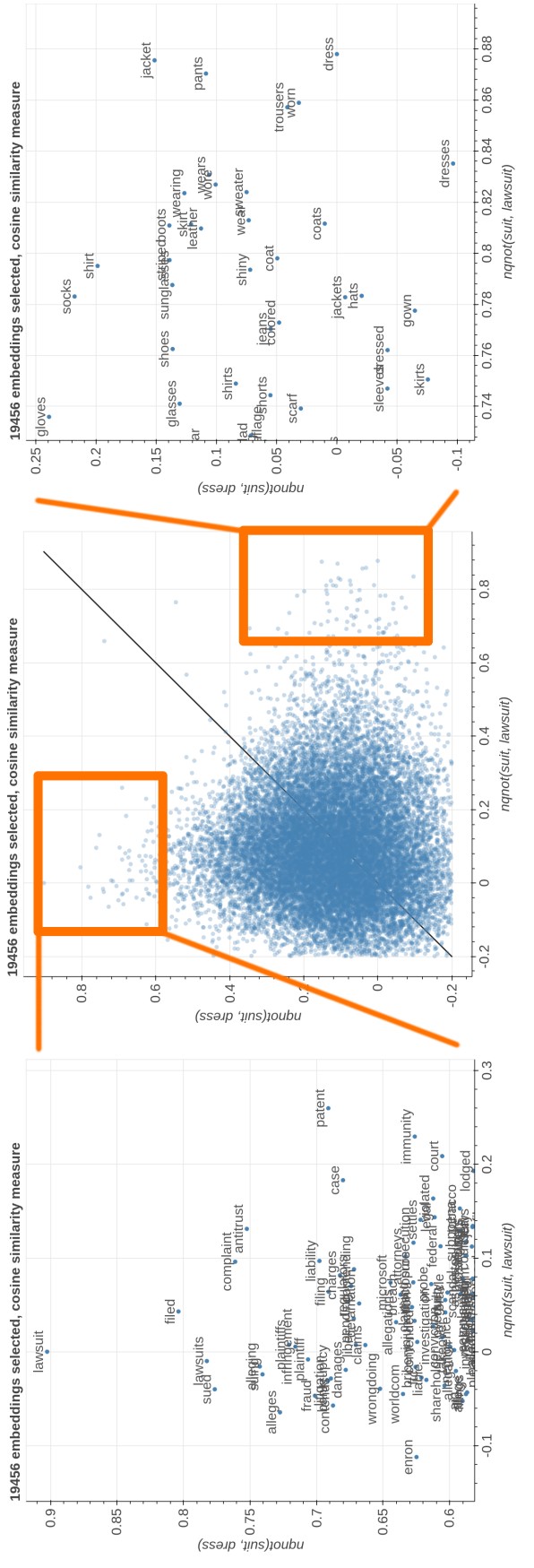

Figure 8: Plot of embeddings in *Wikipedia* with *suit* negated with respect to *lawsuit* and *dress* respectively as axes.

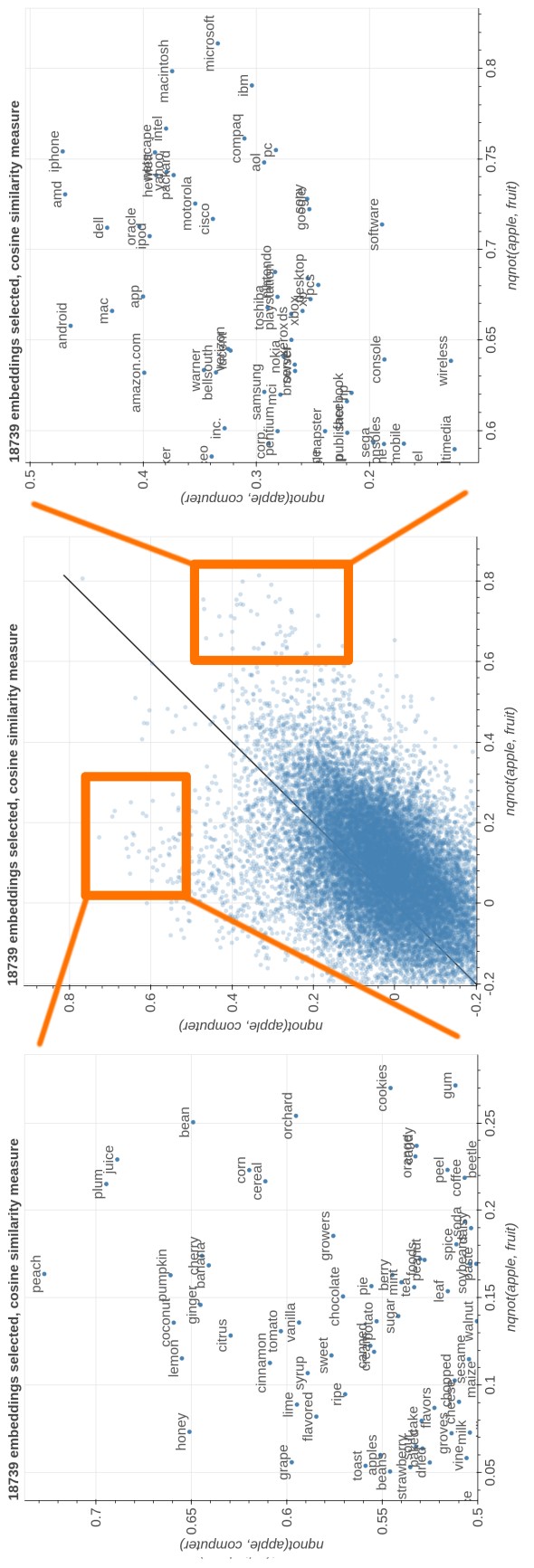

Figure 9: Plot of embeddings in *Wikipedia* with *apple* negated with respect to *fruit* and *computer* respectively as axes.

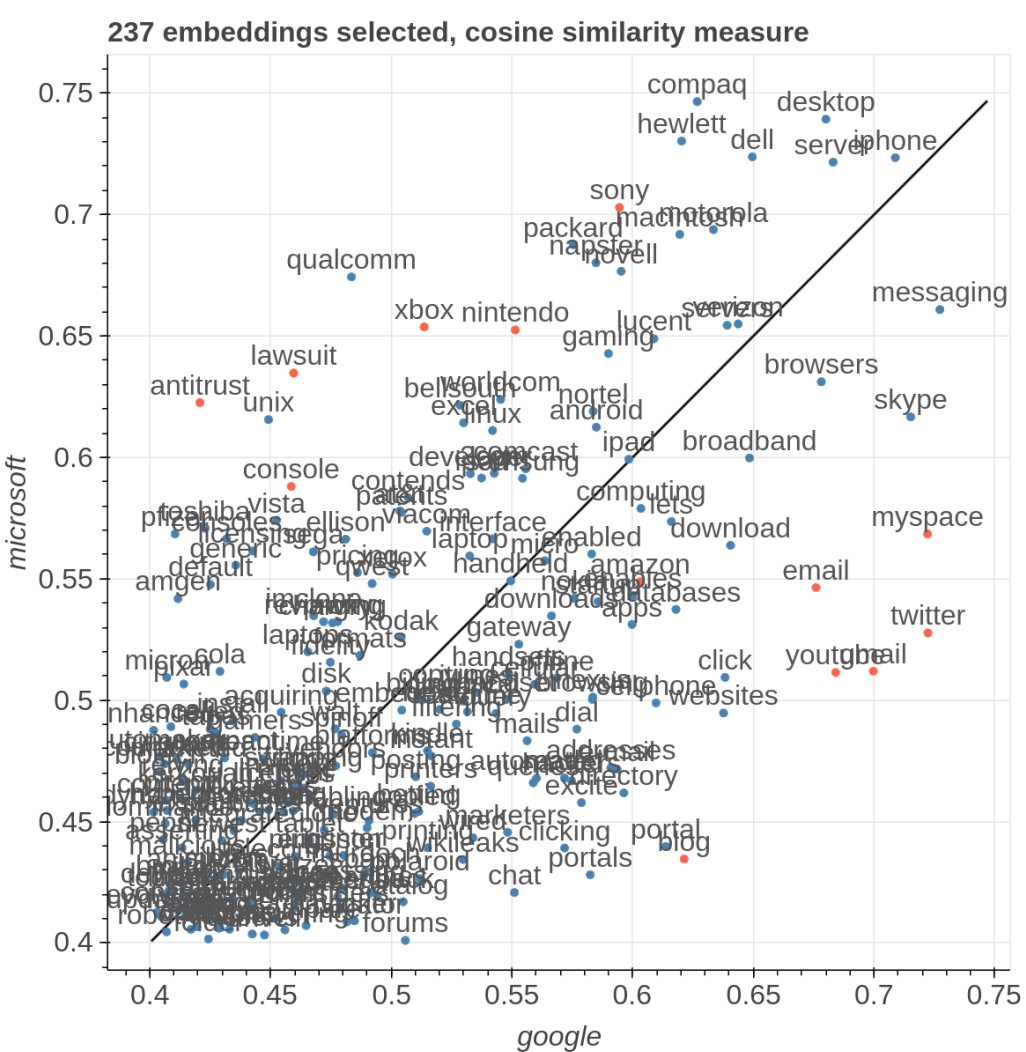

Figure 10: Fine-grained comparison of the subspace on the axis *google* and *microsoft* in *Wikipedia*.

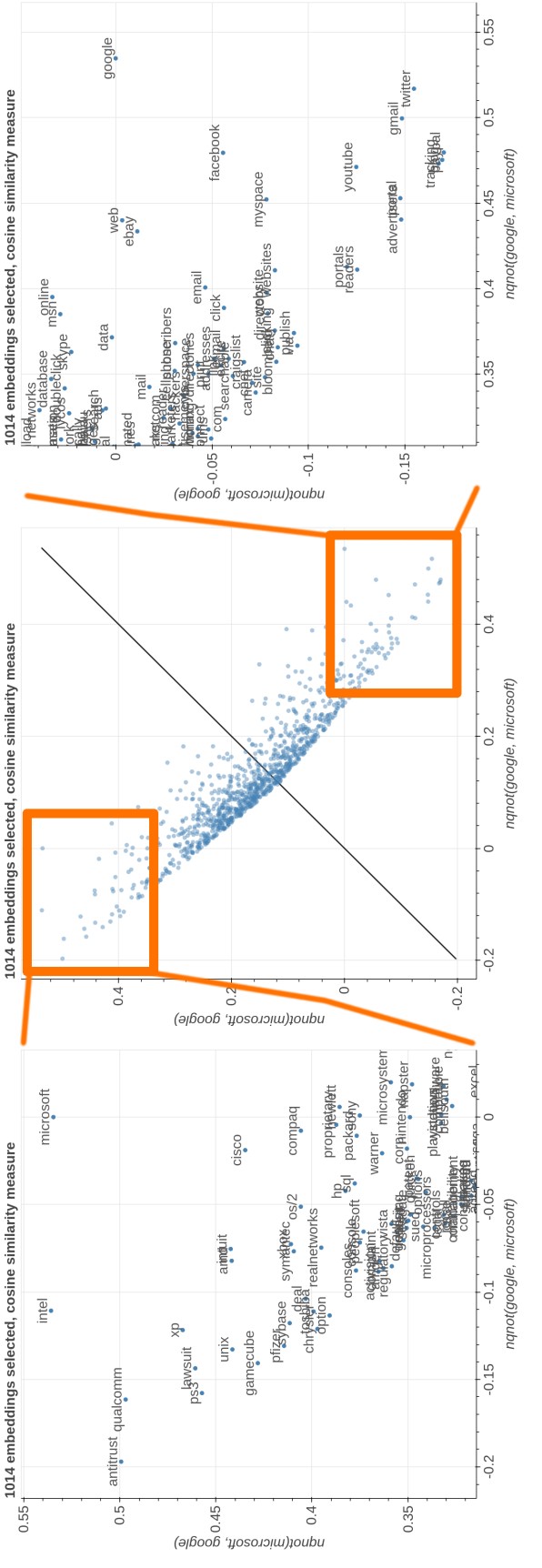

Figure 11: Fine-grained comparison of the subspace on the axis $nqnot(google, microsoft)$ and $nqnot(microsoft, google)$ in *Wikipedia*.

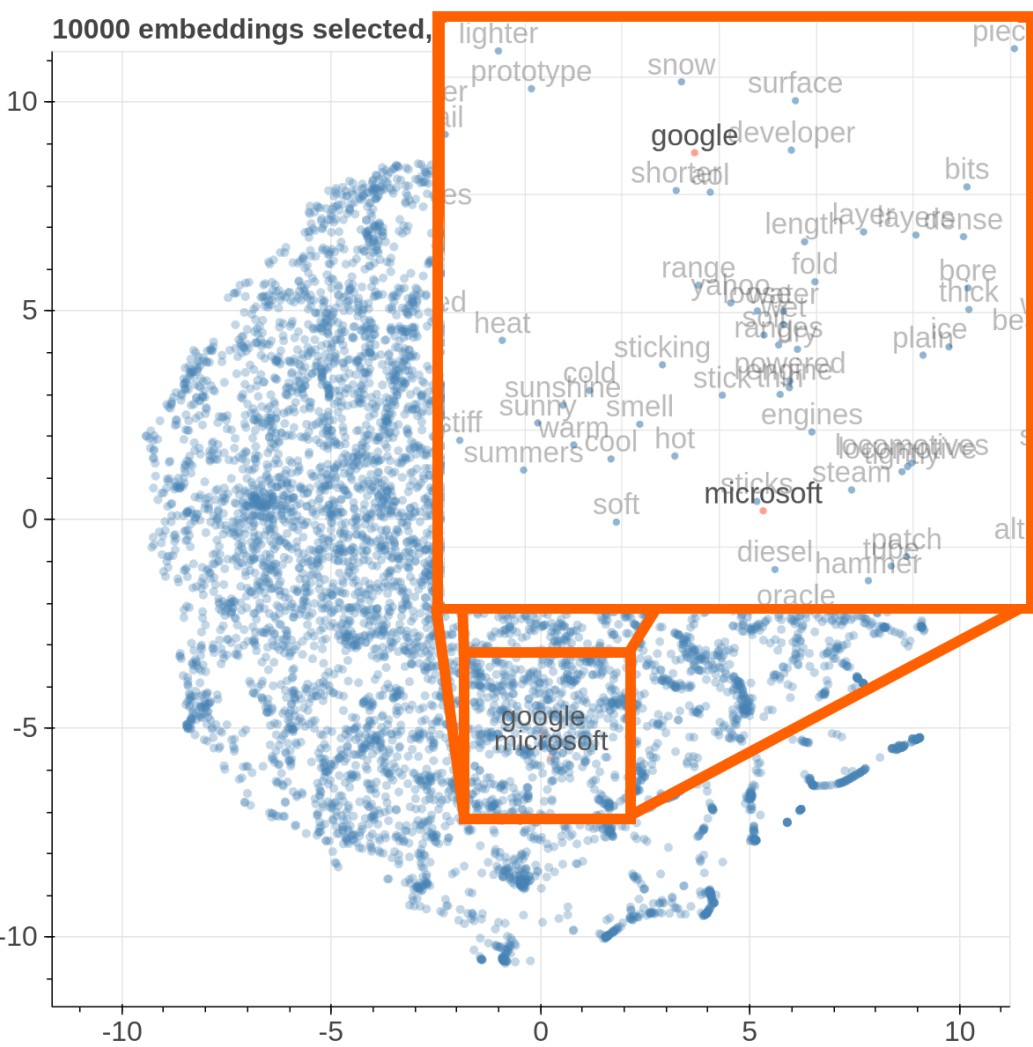

Figure 12: t-SNE visualization of *google* and *microsoft* in *Wikipedia*.

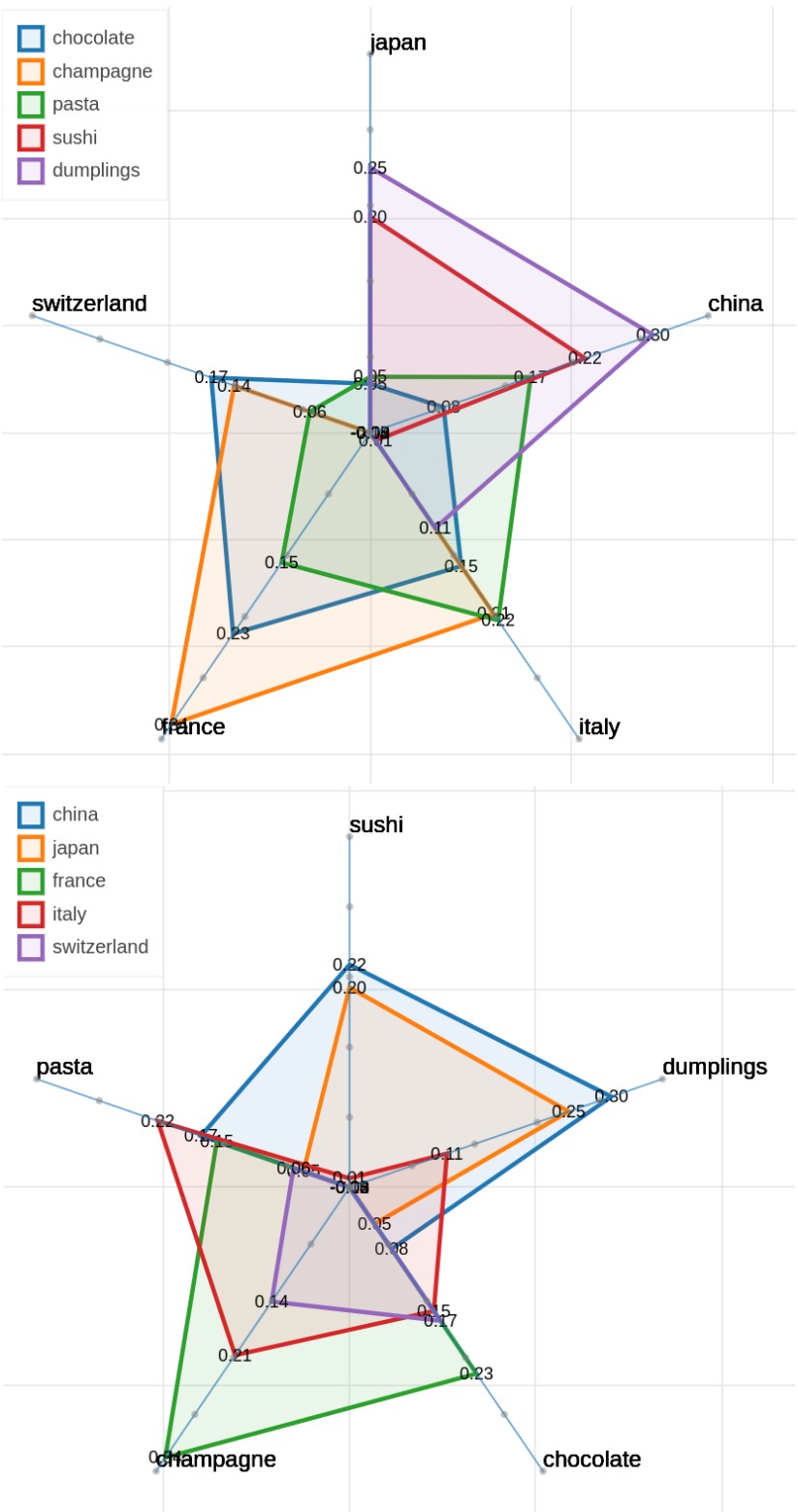

Figure 13: Two polar view of countries and foods in *Wikipedia*.

