# OpenReview forum: "Visualizing and Understanding the Semantics of Embedding Spaces via Algebraic Formulae"
_ICLR.cc/2019/Conference_

### Official Review · AnonReviewer1 · 2018-11-02
**interesting but not clear how useful**

**Rating:** 4
**Confidence:** 3

**Review:**

To the best of my understanding the paper proposes some methodological ideas for visualizing and analyzing representations.
The paper is unclear mainly because it is a bit difficult to pinpoint the contribution and its audience. What would help me better understand and potentially raise my rating is an analysis of a classical model on a known dataset as a case study and some interesting findings would help make it more exciting and give the readers more incentives to try this out. Like train an AlexNet and VGG imagenet model and show that the embeddings are better aligned with the wordnet taxonomy in one of the other. This should be possible with their approach if i understand it correctly.

pros:
- visualization and analysis is a very exciting and important topic in machine learning
- this is clearly useful if it worked
cons:
- not sure what the contribution claim for the paper is since these types of plots existed already in the literature (is it a popularization claim ?)

---

> ### Author Response · Authors · 2018-11-13
> **rebuttal**
>
> Thank you for your comments and please see the inlined answers regarding your concerns.
>
> >>> To the best of my understanding the paper proposes some methodological ideas for visualizing and analyzing representations. The paper is unclear mainly because it is a bit difficult to pinpoint the contribution and its audience. What would help me better understand and potentially raise my rating is an analysis of a classical model on a known dataset as a case study and some interesting findings would help make it more exciting and give the readers more incentives to try this out. Like train an AlexNet and VGG imagenet model and show that the embeddings are better aligned with the wordnet taxonomy in one of the other. This should be possible with their approach if i understand it correctly.
>
> We did perform an analysis of a classical model (GloVe) on a known dataset (Gigaword) and reported several case studies in the Case Studies section reporting interesting findings:
> the presence of gender bias in different datasets and how it changes over the different datasets,
> the characterization of fine-grained differences between extremely close vectors, showing how the embeddings encode polysemy by showing a plane on which multiple senses of words are separable.
> It seems to us we already did what the reviewer was asking us to do, but just on a purely linguistic dataset.
>
> Also, our work is not related with wordnet taxonomies. It would be appreciated if the reviewer could kindly point us to how we should proceed in aligning the embeddings and the wordnet taxonomies, as it is not clear to us and seems unrelated to our work.
>
> >>> pros:
> 	- visualization and analysis is a very exciting and important topic in machine learning
> 	- this is clearly useful if it worked
>
> The combination of case studies and user study that we reported in the paper suggest that it actually works.
>
> >>> cons:
>         - not sure what the contribution claim for the paper is since these types of plots existed already in the literature (is it a popularization claim ?)
>
> We find it hard to derive actionable items from this comment. We ask the reviewer to kindly provide references where plots using explicit formulae as axes are used. To the best of our knowledge, we are the first to propose such a methodology.

---

### Official Review · AnonReviewer2 · 2018-11-02

**Rating:** 3
**Confidence:** 3

**Review:**

The idea of analyzing embedding spaces in a non-parametric (example-based) way is well-motivated. However, the main technical contribution of this paper is otherwise not clear - the methodology section covers a very broad set of techniques but doesn't provide a clear picture of what is novel; furthermore, it makes a strong assumption about linear structure in the embedding space that may not hold. (It's worth noting that t-SNE does not make this assumption.)

The visualization strategies presented don't appear to be particularly novel. In particular, projection onto a linear subspace defined by particular attributes was done in the original word2vec and GloVe papers for the analogy task. There's also a lot of other literature on interpreting deeper models using locally-linear predictors, see for example LIME (Ribeiro et al. 2016) or TCAV (Kim at el. 2018).

Evaluations are exclusively qualitative, which is disappointing because there are quantitative ways of evaluating a projection - for example, how well do the reduced dimensions predict a particular attribute relative to the entire vector. Five-axis polar plots can pack in more information than a 2-dimensional plot in some ways, but quickly become cluttered. The authors might consider using heatmaps or bar plots, as are commonly used elsewhere in the literature (e.g. for visualizing activation maps or attention vectors).

User study is hard to evaluate. What were the specific formulae used in the comparison? Did subjects just see a list of nearest-neighbors, or did they see the 2D projection? If the latter, I'd imagine it would be easy to tell which was the t-SNE plot, since most researchers are familiar with how these look.

---

> ### Author Response · Authors · 2018-11-13
> **rebuttal part 2**
>
> >>> Evaluations are exclusively qualitative, which is disappointing because there are quantitative ways of evaluating a projection - for example, how well do the reduced dimensions predict a particular attribute relative to the entire vector.
>
> Yes, there are quantitative measures for comparing different algorithmic ways to perform dimensionality reduction. However, what we proposed is not an algorithm to perform dimensionality reduction, but a methodology to support users to encode intentions (concepts she cares about) explicitly to the axes of the projection. Such intention is different case by case in analytical tasks. The attributes the user cares about are explicitly encoded in the algebraic formulae she decides to use, so using the reduced dimensions to predict the attributes is meaningless.
>
> We made it mode clear in the first paragraph of the evaluation section that we are not trying to find an optimal dimensionality reduction technique, but comapring an approach to keep users in the loop:
> “The goal is to find out if and how visualizations using user-defined semantically meaningful algebraic formulae as their axes help users achieve their analysis goals.
> What we are not testing for is the quality of projection itself, as in PCA AND t-SNE the projection axes are obtained algorithmically, while in our case they are explicitly defined by the user.”
>
> >>> Five-axis polar plots can pack in more information than a 2-dimensional plot in some ways, but quickly become cluttered. The authors might consider using heatmaps or bar plots, as are commonly used elsewhere in the literature (e.g. for visualizing activation maps or attention vectors).
>
> We explicitly explained in the 3rd paragraph that the polar view is to be preferred in the case where the analysis goal needs more than 2 dimensions of variability, but where the number of items is limited for the reason that several items would make the visualization cluttered. The problem with bar plots and heatmaps is that themselves they don't scale either to a big number of axes of elements and they rely on hue (in the case of heatmaps) and difference in size (in the case of barplots) to provide the information, while the polar view relies on several visual variables at the same time (position, dimension, hue, shape) for the same task, making it a better choice.
>
> >>> User study is hard to evaluate. What were the specific formulae used in the comparison? Did subjects just see a list of nearest-neighbors, or did they see the 2D projection? If the latter, I'd imagine it would be easy to tell which was the t-SNE plot, since most researchers are familiar with how these look.
>
> The tasks and the formulae we used were [banana & strawberry, google & microsoft, nerd & geek, book & magazine, 110392 & 95212, 387862 & 42956, 278209 & 230444, 162363 & 307542], the numbers are the same terms but obfuscated, as described in the manuscript. We added them in the description of the experiment.
>
> As we are evaluating visualizations, the users are shown visualizations of the 2d projections as described in the second paragraph of the user study section.
> They are told which visualizations are t-SNE and which use explicit axes as they are required to express a preference at the end (which is reported). Knowing which plot they were looking at was also needed in order to provide an interpretation to the axes (in the t-SNE they are meaningless, without labels, in the explicit case the axes report the formula used to obtain them in order to be interpretable).
> Can you elaborate on why do you believe this is a problem?

---

> ### Author Response · Authors · 2018-11-13
> **rebuttal part 1**
>
> Thank you for your comments and please see the inlined answers regarding your concerns.
>
> >>> However, the main technical contribution of this paper is otherwise not clear - the methodology section covers a very broad set of techniques but doesn't provide a clear picture of what is novel;
>
> We axplcitly outlined the contribution in the revised manuscript.
> What we described in the methodology is how to map an analysis task in terms of the items to visualize and the dimensions to visualize them on (again, defined as explicit formulae). To the best of our knowledge, there was no precedent work and ours is entirely novel.
>
> >>> furthermore, it makes a strong assumption about linear structure in the embedding space that may not hold. (It's worth noting that t-SNE does not make this assumption.)
>
> The proposed methodology doesn't make any assumption on the structure of the embedding space itself, regardless of whether it is linear or not. What we proposed is to slice the space with a hyperplane that is semantically meaningful with respect to the analysis goal of the user. The slicing can is linear (high-dimensional manifold in case of polar view) but we don’t make assumptions embedding space.
>
> >>> The visualization strategies presented don't appear to be particularly novel. In particular, projection onto a linear subspace defined by particular attributes was done in the original word2vec and GloVe papers for the analogy task.
>
> We find it difficult to relate with this comment and would like to ask the reviewer to kindly provide more details on specific papers and figures.
>
> In the original word2vec paper (Distributed Representations of Words and Phrases and their Compositionality) there is one 2d PCA projection, while in the original GloVe paper (GloVe: Global Vectors for Word Representation) there is no visualization or projection. If the reviewer were referring to the examples on the GloVe website (https://nlp.stanford.edu/projects/glove/) those are again 2d PCA projections with no interpretable semantics along the axes. Or, if Figure 2 in Linguistic Regularities in Continuous Space Word Representations was referenced, that is a cartoon image.
>
> >>> There's also a lot of other literature on interpreting deeper models using locally-linear predictors, see for example LIME (Ribeiro et al. 2016) or TCAV (Kim at el. 2018).
>
> We are aware of this line of work using local descriptors for interpreting deep models, but find it less relevant to the analysis and visualization of embeddings. It would be helpful if the reviewer could help clarify the concerns as local interpretability is an approach applicable to various problem domains, but we don’t see how it is related to to our proposal as we are not learning any predictor.

---

### Official Review · AnonReviewer3 · 2018-11-06
**ideas seems a common practice in various prior visualization tasks**

**Rating:** 3
**Confidence:** 4

**Review:**

Paper presented a new and simple method to visualize the embedding space geometry rather than standard t-SNE or PCA. The key is to carefully select items to be visualized/embed and interpretable dimensions. A few case study and user study were conducted to show the benefit of the proposed approach.

- on algebraic formulae (AF): it would be good to clarify the def of AF explicitly. Rules/extention/axes are not very clear and mathematically consistent in section 3. Would projection idea be applied to arbitrary AFs?

- while the idea being simple, I am not quite confident about the novelty. For example for the de-bias application, Bolukbasi et al. had already did the same plot along the he-she axis. Similar plots on the polysemous word embedding can be found in Arora et al., 2017, etc.

- The user study with n=10 are typically less reliable for any p-value evaluation.

---

> ### Author Response · Authors · 2018-11-13
> **No previous visualization actually used this simple idea**
>
> Thank you for your comments and please see the inlined answers regarding your concerns.
>
> >>> - on algebraic formulae (AF): it would be good to clarify the def of AF explicitly. Rules/extention/axes are not very clear and mathematically consistent in section 3. Would projection idea be applied to arbitrary AFs?
>
> Yes, we allow users to create AFs by compositing vector math operators (e.g., add, sub, mul) to be applied on vectors (referenced by their label in the data, i.e. the vector(apple) is obtained by using the keyword “apple” in the formula). For instance, “(he + him) /2”, resolves he and him with their respective vectors, sums them and then divides the resulting vector by two. We added the following sentence to the manuscrip in the methodology sectiont:
> Algebraic formulae are a composition vector math operators (e.g., add, sub, mul) to be applied on vectors (referenced by their label in the data, i.e. the vector of ``apple'' is obtained by using using the keyword ``apple'' in the formula)..
>
> >>> - while the idea being simple, I am not quite confident about the novelty. For example for the de-bias application, Bolukbasi et al. had already did the same plot along the he-she axis. Similar plots on the polysemous word embedding can be found in Arora et al., 2017, etc.
>
> We acknowledge that those works are relevant, but argue that the information provided in their plots and the nature of their plots is different from our proposal.
>
> In Bolukbasi et al., the x-axis is the difference between he and she, while the y-axis is a learned direction encoding neutrality (through SVD). As such, only one of the axes is explicitly defined by an algebraic formula. The example in Bolukbasi et al. indeed is a subset the methodology that we are proposing, which is more generic and can be applied widely.
>
> Regarding Arora et al. the task may be similar, but the plots are obtained through isometric mapping rather than explicitly-defined semantically-meaningful subspaces. That is, axes in their plots are not interpretable, which makes their approach more closer to PCA or t-SNE projections and has little in common with our proposal.
>
> >>> - The user study with n=10 are typically less reliable for any p-value evaluation.
>
> While we agree more test subjects usually lead to (perceptually) better statistical confidences, the ANOVA and t-test results in our study already suggested significant difference (p values being magnitudes smaller than 0.01, as described in the result analysis section).

---

> > ### Comment · AnonReviewer3 · 2018-11-21
> > **don't agree with some response.**
> >
> > I don't agree with author on their novelty with the response. For instance, Figure 4 in https://arxiv.org/pdf/1607.06520v1.pdf conveys similar ideas of using the proposed approach. [Indeed the same plot as Figure 1 in this paper]
> >
> > Actually, if just simply google serach, you will hit this following paper:
> > https://web.stanford.edu/class/cs224n/reports/6835575.pdf
> > with the exact plots as the author may argue.
> >
> > Further, as the author mentioned in literature,
> > http://embvis.flovis.net/
> > http://embvis.flovis.net/s/scatter.html
> > where you can get the exact plots with single concept or average of multiple concept to project.

---

> > > ### Author Response · Authors · 2018-11-22
> > > **rebuttal for each of the suggested visualizations**
> > >
> > > Thank you for providing this additional references, we'll add them as relevant literature.
> > >
> > > Regarding the first one (Figure 4 of Bolukbasi et Al.) it surely is relevant, but as you can see the axis are the difference of two embeddings in two different embedding spaces. This makes the plot difficult to interpret as it shows only how different, for each word represented by a dot in the scatter plot it is more biased in one or the other dataset / embedding method, but the reader has no idea of how much the word is biased to begin with. In contrast, in our comparison view, you can clearly se the position of the word in both datasets and a line is drawn between them so that the reader can understand both the absolute degree of bias of each word in each dataset and how much the dataset differ in terms of bias.
> > >
> > > Regarding the second one, it doesn't seem to be a published paper, but rather a report that a student put on their website, so it doesn't seem like something we would be able to cite. In any case, as much as that work is relevant, in all the figures the author provide a side by side view of the embeddings projection in the two datasets, which makes it really hard to compare the relative location, and how much delta there is between the two locations of the same embedding in the different embedding spaces. Thus we believe that our comparison view provides a better visualization.
> > >
> > > Finally, regarding Heimerl et al., we already describe the difference in the related section, but we can add that our approach has several advantages compared to that tool: first the user can define any algebraic formula as axes rather than just averages of embeddings, and fo instance the fine grained analysis plots and the polysemy plot we provide in the paper would not be obtainable; second: as you can define averages as axes, our approach subsume their, being more general; third we also provide the comparison view for cases where the user wants to compare different embedding spaces and the polar view where the goal requires more than two axes, both things are not provided by Heimerl et al.
> > >
> > > Finally, as a general remark, we are proposing to use this simple idea as a general methodology, while most of the plots you refer to (a part from Heimerl's tool) are spurious specific plots obtained in specific circumstances, so the fact that someone used a similar plot to what is obtainable with our methodology is just an additional example use case that strengthens even more the usefulness of the methodology.

---

### Meta-Review · Area_Chair1 · 2018-12-17
**All reviewers agree that paper is not strong enough**

**Confidence:** 4
**Recommendation:** Reject

**Metareview:**

Several visualizations are shown in this paper but it is unclear if they are novel.